# The insidious degeneration of white matter and cognitive decline in Fabry disease

Jacob W. Johnson[1], Hediyeh Baradaran[2], Jubel Morgan[2], Henrik Odèen [2], Emma Friel[3], Carrie Bailey[3], Brandon A. Zielinski[2,4], Hunter R. Underhill [2,3,5]*

1 Department of Biomedical Engineering, University of Utah, Salt Lake City, Utah, United States of America, 2 Department of Radiology and Imaging Sciences, University of Utah, Salt Lake City, Utah, United States of America, 3 Department of Pediatrics, Division of Medical Genetics, University of Utah, Salt Lake City, Utah, United States of America, 4 Department of Pediatrics, Division of Neurology, University of Utah, Salt Lake City, Utah, United States of America, 5 Huntsman Cancer Institute, University of Utah, Salt Lake City, Utah, United States of America

* Hunter.Underhill@hsc.utah.edu

## Abstract

Fabry disease is a rare X-linked deficiency of lysosomal alpha-galactosidase that causes glycolipid accumulation in tissues, including the brain. The most common neurologic sequelae of Fabry are cognitive decline and white matter lesions (WMLs) on brain magnetic resonance imaging (MRI). In the at-large population, however, WMLs are non-specific, highly prevalent, and most are clinically silent. Thus, we compared Fabry to typical brain aging to identify factors unique to Fabry-related cognitive decline. Twenty adult Fabry patients (75% female; median age 36.4 yrs, range: 19.8–63.2 yrs; 95% on enzyme replacement therapy) without a history of stroke or other neurologic diseases and 20 age/sex-matched healthy controls were enrolled in a case-control study. All participants underwent a neurocognitive assessment and a 3.0 T MRI study of the brain that used structural MRI (e.g., fluid-attenuated inversion recovery, FLAIR), semi-quantitative MRI (e.g., normalized FLAIR signal intensity), and quantitative MRI (diffusion tensor imaging, bound-pool fraction imaging). During a blinded review of structural MRIs, a neuroradiologist's categorization of case-control status did not correspond to disease status (Fisher's test, $P > 0.99$) but rather to age ($P = 0.004$), indicating qualitative changes associated with Fabry were similar to normal age-related brain alterations. Using quantitative MRI, however, we detected evidence of microstructural damage in the white matter of younger Fabry adults (<40 yrs). With age, WML severity increased and the corpus callosum atrophied in Fabry, phenomena absent in controls and consistent with progressive tissue damage. Neurocognitive assessments identified trends for lower verbal intelligence quotient and executive function in the younger Fabry participants, which became statistically significant in the older Fabry patients. Our data suggest that the early onset of microstructural damage in Fabry drives the insidious degeneration of white matter, leading

**Data availability statement:** All data not contained in the paper necessary to replicate plots and figures are available in the S1 File. All magnetic resonance images (DICOM, RAW) used in the study are available from the Dryad Data Repository (https://doi.org/10.5061/dryad. gb5mkkx3g).

**Funding:** Genzyme, a Sanofi corporation supported this investigator sponsored study (GZ-2015-11321). The MRI resources used were partially funded by an NIH Shared Instrumentation Grant (S10OD018482).

**Competing interests:** Genzyme, a Sanofi corporation, provided salary support to EF, BAZ, and HRU under the investigator sponsored study agreement. The conduct of the study, acquisition of data, data analysis, and manuscript were controlled by the study authors. No authors identified additional conflicts of interest.

to impaired cognition. Aging Fabry patients may benefit from serial cognitive assessments to identify unmet therapeutic needs.

## Introduction

Fabry disease is a rare X-linked disorder caused by a deficiency of the lysosomal enzyme alpha-galactosidase [1]. Alpha-galactosidase, encoded by the *GLA* gene, catalyzes the removal of terminal alpha-galactose from glycoproteins and glycolipids [2]. The reduced activity of alpha-galactosidase in Fabry leads to the accumulation of the principal lysosomal substrate globotriaosylceramide (lyso-$Gb_3$) in tissues. Lyso-$Gb_3$ deposition occurs in endothelial cells, epithelial cells, pericytes, myocardial cells, ganglion cells, and smooth muscle cells causing a multi-organ, systemic disease [3]. In the central nervous system (CNS), sequelae of Fabry manifest as stroke, non-specific white matter lesions (WMLs), and cognitive decline. In a review of 2,446 patients from the Genzyme Fabry Disease Registry, stroke was documented in 6.9% of males and 4.3% of females [4]. The risk of stroke before 45 years of age was 12-fold higher for males and >4-fold higher for females compared to the corresponding age and sex-matched general population [4]. WMLs, focal macrostructural hyperintensities identified on brain MRI (i.e., leukoaraiosis), are present in 42–81% of patients with Fabry [5–7]. In Fabry, WMLs are identified in 15.9% of children [8], increase in size and abundance with age [9], and occur with a similar prevalence in males and females [9,10]. Neuropsychiatric studies of Fabry have described deficits in attention [11,12], executive function [6,12–15], and information processing speed [6,12–15]. Although enzyme replacement therapy (ERT) was introduced in 2001 for the treatment of Fabry [16], the CNS sequelae associated with Fabry have persisted [17–19].

The etiology of cognitive changes in Fabry is unclear because cognitive decline occurs in patients with or without a history of stroke. The high prevalence of WMLs in Fabry has suggested a possible association with neuropsychiatric findings, but comparisons have yielded mixed results. Schermuly et al. studied 25 Fabry patients and reported the volume of WMLs was similar between Fabry and 20 age, gender, and education-matched controls [11]. Although WML volume in Fabry was inversely correlated with total verbal learning and visual memory, accounting for age extinguished these associations [11]. In contrast, Ulivi et al. found a significant increase in WML volume in 31 Fabry patients, three of whom had a history of stroke or transient ischemic attack (TIA), compared to 19 healthy controls matched for age and sex [6]. A strong inverse correlation between WML volume and processing speed was observed in Fabry [6]. Murphy et al. compared 26 patients with Fabry to 18 healthy controls matched for age and premorbid intellectual level and found WML volume in Fabry to be significantly increased [15]. Significant inverse associations were found in Fabry between WML volume and both performance intelligence quotient (IQ) and information processing speed [15]. The varied results of these studies are difficult to collectively interpret, particularly when the routine occurrence of WMLs in the general

population is considered. Periventricular and subcortical WMLs are increasingly prevalent with age and present in 80% and 92% of individuals 60–90 years old, respectively [20]. Although more advanced WMLs have been associated with cognitive impairment [21], isolated WMLs are generally asymptomatic. Thus, the evidence for WMLs as the source of cognitive decline in Fabry remains unclear.

In addition to the macrostructural WMLs present in Fabry, microstructural damage has also been considered a source of cognitive decline. Diffusion tensor imaging (DTI) is a quantitative MRI technique used to measure brain tissue integrity and connectivity. Microstructural changes in white matter detected with DTI precede the development of macrostructural changes in Fabry [22]. A significant reduction in fractional anisotropy (FA) and increased mean diffusivity (MD), a combination indicative of damaged or reduced myelin, has been observed in brain regions of Fabry patients compared to healthy controls [23,24]. Paavilainen et al. used a whole-brain, voxel-based analysis to show a widespread reduction of FA and increased MD [23]. In a similar analysis of 31 Fabry patients, Ulivi et al. showed significantly reduced FA and increased MD in bilateral supratentorial white matter regions compared to 19 age/sex-matched controls [6]. In addition, a negative correlation between mean white matter FA and processing speed was described. The implications of reduced FA and increased MD were strengthened by a report showing decreased structural connectomes in Fabry [25]. Although global alterations in diffusion-based parameters have been recognized in Fabry, the findings are non-specific to Fabry and similarly occur during normal brain aging [26,27].

In this study, we sought to build upon prior investigations by further evaluating associations between cognitive decline, WMLs, and connectivity in a stroke-free Fabry cohort. Healthy controls matched to the age/sex of each Fabry patient were identically studied. Neuropsychiatric tests were obtained in parallel with MRI. Qualitative images (FLAIR, T2W, and MP-RAGE) were used to compare macroscopic differences between cohorts. Semi-quantitative images (e.g., normalized FLAIR signal intensity) and quantitative images (bound-pool fraction imaging) were used to detect microscopic changes in Fabry patients. Bound-pool fraction imaging measures the fraction of hydrogen molecules bound to macromolecules such as myelin and has been previously applied to the study of myelin density [28], Fabry [24], multiple sclerosis [29], and neuropsychiatric disorders [30]. Finally, DTI was used to isolate and evaluate connectivity and myelination of the corpus callosum. We studied the corpus callosum because (1) this fiber bundle is the major commissural pathway connecting the brain hemispheres and provides a robust target for the detection of potentially subtle perturbations in diffusion-based metrics [31], (2) WMLs in the corpus callosum are uncommon, including in Fabry, thereby minimizing confounding effects [32,33], and (3) prior whole-brain, voxel-based studies have found evidence of altered FA and MD in the corpus callosum of Fabry patients [6,23]. Subsequently, we compared imaging metrics to age and neuropsychiatric parameters in Fabry and controls.

## Materials and methods

### Participant group

Adult (≥18 years) males and females with Fabry disease were recruited from the University of Utah Metabolic Clinic and mailers were sent to patients enrolled in the Genzyme Fabry Disease Registry. Fabry patients with a history of stroke, neurologic disease (e.g., multiple sclerosis), and cardiac pacemakers or other implantable devices were excluded from enrollment. A control group of healthy male and female volunteers with no history of neurologic disease (e.g., stroke, brain tumor, other known neurodegenerative diseases, etc.) was recruited during the study period to closely match the age and sex of each Fabry participant. The study was approved by the University of Utah Institutional Review Board (#IRB_00092478). All participants gave their written informed consent.

### Health questionnaires and neurocognitive assessment

All participants completed the following three questionnaires: 1) General Health (demographics, history of high blood pressure, high cholesterol, etc.; nine questions), 2) Center for Epidemiological Studies Depression Scale (CES-D), and 3)

RAND 36-Item Health Survey. All participants completed the following four neurocognitive evaluations: 1) Wechsler Adult Intelligence Scale, Third Edition (WAIS-III), 2) Conners' Continuous Performance Test II (CPT-II), 3) NIH Toolbox assessments – Pattern Comparison Processing Speed and Flanker Inhibitory Control and Attention Test, and 4) Trail Making Test – test of visual attention and task switching. The questionnaires and neurocognitive evaluations occurred within two weeks of the participant's corresponding MRI, were conducted in person by the study nurse with >20 years of experience administering neuropsychological assessments, and the assessment duration was < 4 hours.

## Image acquisition

Images were acquired on a 3.0 Tesla MRI scanner (Prisma, Siemens, Erlangen, Germany) using a 64-channel receive-only head coil (Siemens) at the University of Utah. All images were acquired in the axial plane without angulation.

Three qualitative image series were acquired: T2-FLAIR (fluid-attenuated inversion recovery, T2W (T2-weighted), and MP-RAGE (magnetization-prepared rapid acquisition gradient echo). FLAIR images used a three-dimensional (3D), spoiled, fast low angle shot sequence (FLASH; repetition time/echo time [TR/TE]=6,000/392 ms, time to inversion [TI]=2,100 ms) with a 256x256 field-of-view (FOV), 2.0 mm slice thickness, 1.0x1.0x2.0 mm$^3$ acquisition resolution (60 slices), one signal average, and a generalized autocalibrating partially parallel acquisition (GRAPPA) [34] factor of 2 in the phase encode (PE) direction. Acquisition time was 4 min 44 sec. T2W images used a 3D, spoiled, FLASH (TR/TE = 3,200/564 ms) with a 256x256 FOV, 2.0 mm slice thickness, 1.0x1.0x2.0 mm$^3$ acquisition resolution (96 slices), and GRAPPA = 2 (PE). Acquisition time was 2 min 26 sec. MP-RAGE images used a 3D, T1-weighted gradient echo sequence (TR/TE = 2,000/2.19 ms, $\alpha$ = 8°, TI = 1,100 ms) with a 256x256 FOV, 1.0 mm slice thickness, 1.0x1.0x1.0 mm$^3$ acquisition resolution (160 slices), one signal average, 0.875 phase partial Fourier, and GRAPPA factor of 2 (PE). Acquisition time was 4 min 8 sec.

Bound-pool fraction maps were obtained using a 3D spoiled gradient echo magnetization transfer (MT) sequence (TR/TE = 31/4.6 ms, $\alpha$ = 10°) using a saturation pulse (13 ms duration, 500° nominal effective flip angle) to acquire three Z-spectroscopic data points with offset frequencies ($\Delta$) of 4, 8, and 96 kHz. Acquisition time was 3 min 1 sec for each offset frequency. The variable flip angle (VFA) method used to construct $R_1$ maps used a 3D spoiled gradient echo sequence (TR/TE = 20/4.6 ms) with $\alpha$ = 3°, 10°, and 15°. Acquisition time was 1 min 57 sec for each flip angle. Both the MT and VFA sequences were acquired with a 256x256 FOV, 2.0 mm slice thickness, 2.0x2.0x2.0 mm$^3$ acquisition resolution, one signal average, and 0.75 slice partial Fourier. To correct for $B_0$ heterogeneity, $B_0$ maps were obtained using the dual-TE phase difference method acquired with a 3D spoiled gradient echo (TR/TE$_1$/TE$_2$ = 4.06/1.23/2.46 ms, $\alpha$ = 10°) and 256x256 FOV, 5.0 mm slice thickness, 4.0x4.0x5.0 mm$^3$ acquisition resolution (zero-interpolated to 2.0x2.0x2.0 mm$^3$), one signal average, and acquisition time of 7.2 sec [35]. $B_1$ maps were acquired using the actual flip-angle imaging method with a 3D spoiled gradient echo sequence (TR$_1$/TR$_2$/TE = 25/125/4.6 ms, $\alpha$ = 60°), 256x256 matrix, 4.0x4.0x5.0 mm$^3$ acquisition resolution (zero-interpolated to 2.0x2.0x2.0 mm$^3$), one signal average, and acquisition time of 4 min 36 sec [36].

DTI used a single-shot 2D spin-echo echo-planar sequence with TR/TE = 7,600/78 ms, $\alpha$ = 90°, EPI factor = 128, two b-values of 0 and 1,000 s/mm$^2$, and 64 gradient directions. Images were acquired with a 256x256 FOV, 2.2 mm slice thickness (120 slices), 2.0x2.0x2.2 mm$^3$ acquisition resolution, two signal averages, 0.75 phase partial Fourier, and GRAPPA factor of 2 (PE). Acquisition time was 9 min 32 sec.

## Image processing

Whole-brain bound-pool fraction maps (*f*-maps) were constructed using the matrix model of magnetization transfer for a single parameter determination of *f* as previously described [24,37]. Briefly, $B_1$ maps were used for the voxel-based correction of VFA data during $R_1$ measurements, and both $B_0$ and $B_1$ maps were used for the voxel-based correction of Z-spectroscopic data ($\Delta$ and $\alpha$, respectively). Using $\Delta$ = 96 kHz Z-spectra to normalize the $\Delta$ = 4 and 8 kHz MT images, bound-pool fraction maps were fitted using the corrected $R_1$ maps and Z-spectroscopic data and by applying established

constraints for the four additional cross-relation parameters: $k = 29\ x\ f/(1-f)s^{-1}$, $T_2^F R_1^F = 0.030$, where $F$ = free pool, $B$ = bound pool [24,28]. Whole-brain voxel-based parameter determinations were accomplished using custom-designed software written in Matlab (The Mathworks, Natick, MA) and C/C++.

### Image analysis

A board-certified neuroradiologist with a certificate of added qualification in neuroradiology with over 10 years of experience performed all image review blinded to age and case-control status. During the initial review, the study neuroradiologist was shown FLAIR images in the axial plane from all participants in random order to categorize case-control status. An identical process was performed for T2W and MP-RAGE images. In a separate review, the study neuroradiologist was shown reconstructed images in the sagittal plane for FLAIR, MP-RAGE, and bound-pool fraction maps to identify the centers of the genu and splenium of the corpus callosum. A region of interest (ROI) was also manually drawn in the temporalis muscle on the corresponding qualitative image in the axial plane using a graphical user interface. Greater than one month after the initial image review, FLAIR images were shown to the same neuroradiologist in a new random participant order for (1) determination of presence/absence of WMLs, and (2) Fazekas score assessment, a classification of WML severity [38].

Using the axial plane orthogonal to the sagittal plane, image sets were resampled to isolate supratentorial voxels superior to the plane connecting the genu and splenium of the corpus callosum. A mask specific to each patient and contrast weighting was manually curated and adjusted to ensure the isolation of brain tissues. For FLAIR and MP-RAGE images, intensity was normalized to the mean intensity of the corresponding temporalis muscle ROI. The range for histogram analysis for each image set was based on the distribution of values from all isolated volumes. The range was partitioned into 50 equally-spaced bins. The established range and bins were applied to all samples uniformly. An identical method was performed for bound-pool fraction maps except the data was not normalized to the temporalis muscle. Intensity values for each participant were placed into the appropriate bin, and the bin data for each participant was exported as a proportion of the participant's total number of pixel values in the range.

### DTI analysis

DSI Studio was used to isolate, quantify, and compare the volume of fibers passing through the body of the corpus callosum. The accuracy of b-table orientation was examined by comparing fiber orientations with those of a population-averaged template [39]. The restricted diffusion was quantified using restricted diffusion imaging [40]. The diffusion data were reconstructed using generalized q-sampling imaging with a diffusion sampling length ratio of 1.25 [41]. The tensor metrics were calculated using diffusion-weighted images with a b-value lower than 1750 s/mm².

### Statistical analysis

All statistical tests were performed with GraphPad Prism 10 (version 10.1.0). The independent student's t-test was used for comparisons between groups of continuous data. Pearson's correlation was used to identify associations between parameters. Fisher's exact test was used to examine associations between categorical data. Results were considered statistically significant for $P < 0.05$.

## Results

### Fabry reduces verbal cognition and executive function

Between March 1, 2021 and February 28, 2022, 20 Fabry patients (19 patients on ERT) and 20 age/sex-matched controls were enrolled and underwent neurocognitive assessment and brain MRI. The neurocognitive assessment and MRI were conducted within two weeks of each other for all participants. The demographic and clinical characteristics of Fabry

patients and controls are shown in S1 Fig. By design, no differences in age and sex were observed between cohorts. The absolute mean age difference between case-control pairs was 1.9±1.5 years (Fig 1a). Of the 40 participants enrolled, 24 were younger adults (age<40 yrs). A significant difference in education years was present between Fabry and controls (14.3±1.9 vs. 16.2±1.8 yrs, respectively; $P=0.002$, Fig 1b), which may reflect the recruitment of controls from a university campus.

The CES-D questionnaire did not identify differences in depression between groups (Fisher's test, $P=0.48$; Fig 1c). Well-being, as assessed by the RAND-36 questionnaire, found a statistically significant reduction in physical functioning in Fabry, likely attributable to reduced activity caused by common Fabry symptoms such as heat intolerance secondary to hypohidrosis, increased pain associated with acroparesthesias, or both (Fig 1d).

Compared to controls, a lower IQ driven by a reduced verbal IQ was identified in Fabry (Fig 1e, Table 1). A trend was noted in verbal IQ in the younger cohort for Fabry participants and strengthened into a statistically significant difference in the older cohort. An association between verbal IQ and education years was not detected (Fig 1f). The association between age and verbal IQ was not significant (S2 Fig), but a downward drift with age in both groups was evident and consistent with prior reports on normal aging [42]. Executive function, as measured by error during the Trail Making Test

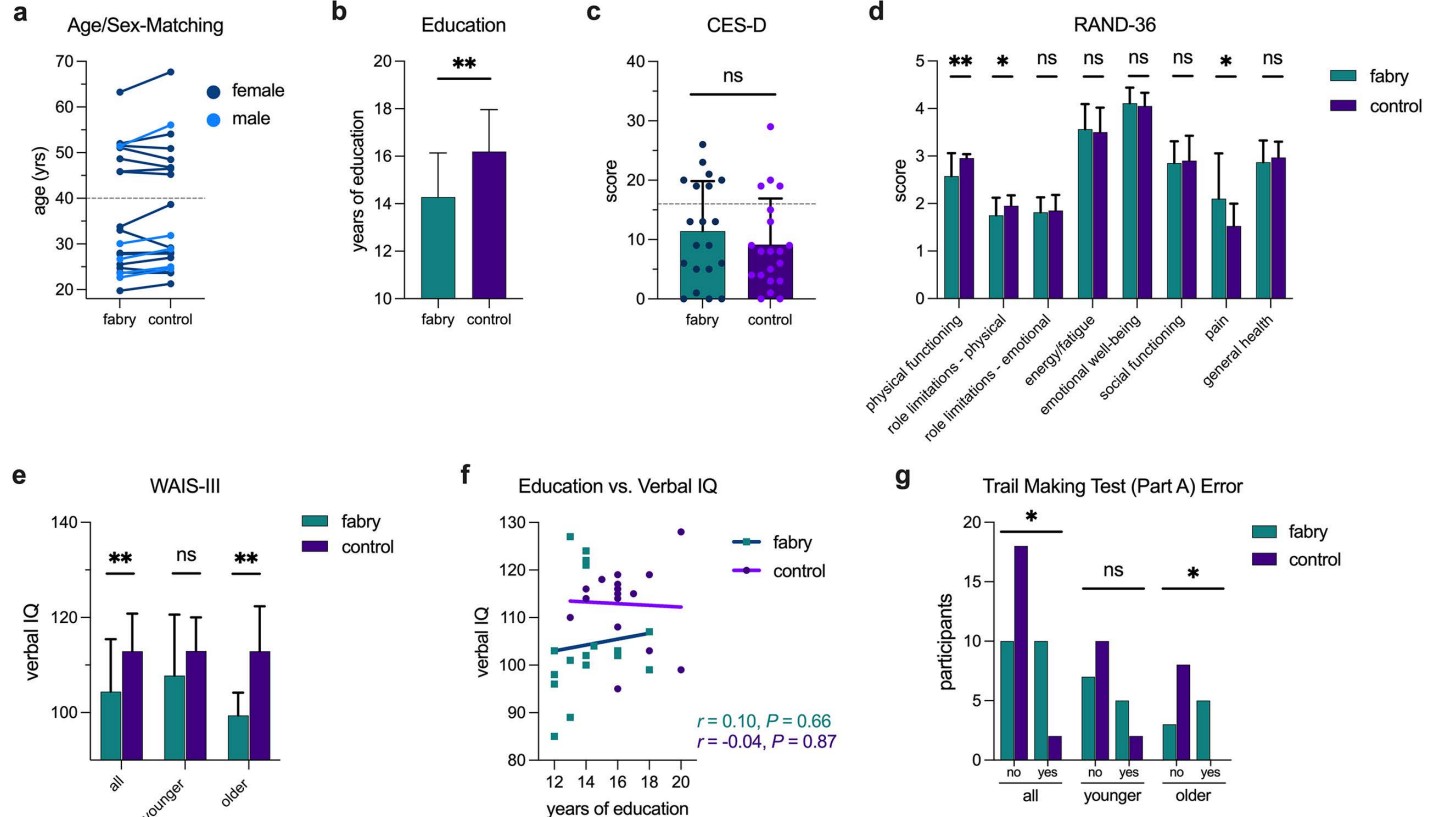

**Fig 1. Fabry disease reduces verbal intelligence quotient (IQ) and executive function.** Each participant with Fabry was closely paired with an age/sex-matched healthy control (a). The dashed line in (a) separates the cohort into younger (<40 yrs) and older (>40 yrs) groups. Although education years differed between cohorts (b), depression as measured by the Center for Epidemiological Studies Depression Scale (CES-D) was similar (c). The dashed line represents the minimum CES-D score to identify a person at risk for clinical depression. Results for the RAND-36 well-being questionnaire are shown in (d). The difference in total IQ using the Wechsler Adult Intelligence Scale, 3rd Edition (WAIS-III) between cases and controls was governed by verbal IQ (e). Verbal IQ was not associated with years of education (f). During the NIH Trail Making Test (Part A), more errors occurred in Fabry (g). Bars indicate mean value and whiskers correspond to standard deviation. ns = not significant, *$P<0.05$, **$P<0.01$, ***$P<0.001$.

**Table 1. Neuropsychiatric performance in Fabry disease compared to healthy controls.**

| Domain | Test | Fabry (*N* = 20) | | Controls (*N* = 20) | | P value |
|---|---|---|---|---|---|---|
| | | Mean | SD | Mean | SD | |
| **Intellectual Function** | **WAIS III, Full Scale IQ** | **107.6** | **10.4** | **114.1** | **8.8** | **0.040** |
| | **WAIS III, Verbal IQ** | **104.4** | **11.0** | **112.9** | **7.9** | **0.008** |
| | WAIS III, Performance IQ | 110.6 | 11.7 | 113.3 | 10.7 | 0.45 |
| **Processing Speed** | TMT, Part A (sec) | 23.2 | 11.4 | 21.0 | 6.5 | 0.45 |
| | TMT, Part B (sec) | 63.0 | 44.7 | 54.3 | 21.3 | 0.44 |
| | WAIS III, Symbol Search | 36.7 | 7.8 | 39.2 | 7.0 | 0.29 |
| | WAIS III, Processing Speed | 107.8 | 13.1 | 112.2 | 11.4 | 0.26 |
| | NIH Toolbox, Oral Symbol Digit Test | 95.6 | 21.5 | 100.2 | 16.1 | 0.44 |
| | **NIH Toolbox, Pattern Comparison[a]** | **91.5** | **18.6** | **107.8** | **20.1** | **0.011** |
| **Attention** | CPT II, Clinical Confidence Index | 61.2 | 9.1 | 65.3 | 11.2 | 0.21 |
| | NIH Toolbox, Flanker Task[a] | 97.4 | 16.3 | 96.4 | 14.6 | 0.85 |
| **Executive Function** | **TMT, Part A, Error** | **0.6** | **0.6** | **0.1** | **0.3** | **0.006** |
| | TMT, Part B, Error | 0.6 | 0.8 | 0.2 | 0.4 | 0.058 |

CPT II, Conners' Continuous Performance Test II; IQ, Intelligence Quotient; NIH, National Institutes of Health; TMT, Trail Making Test; WAIS, Wechsler Adult Intelligence Scale.

[a]age-corrected standard score.

(Part A), was reduced in Fabry (Fig 1g, Table 1). Fabry participants were significantly more likely to make an error during the assessment, a finding that was present as a trend in the younger Fabry cohort and strengthened with age. Years of education (Table 1, S3 Fig) and attention (Table 1) were similar between those with and without errors in the Fabry group. The results for processing speed were conflicting (Table 1). Raw scores from the Trail Making Test indicated a delay in Fabry, which was not supported by trends from the WAIS-III and NIH Toolbox assessments (Table 1). In fact, the age-corrected, standard score from the NIH Toolbox Pattern Comparison showed better processing speed in Fabry (Table 1). Given these inconsistencies, the study may have been underpowered to adequately detect and characterize relatively subtle differences between groups in processing speed, or processing speed was equivocal. Overall, we found evidence for reduced verbal IQ and executive function in Fabry compared to controls which did not appear secondary to depression and education.

## Severity of WMLs in Fabry increases with age

Although we found imaging evidence of increased WML burden in Fabry, particularly in the older cohort (Fig 2), qualitative images were generally similar between cases and controls (Fig 3). Using FLAIR images, the study neuroradiologist was unable to correctly assign case/control status (Fisher's test, $P > 0.99$; S4a Fig). Case-control status was similarly indistinguishable using T2W ($P = 0.75$) and MP-RAGE ($P = 0.52$) images (S4a Fig). During the analysis, however, we observed that image categorization appeared to correspond with age. Thus, the data was reanalyzed to determine if the neuroradiologist's classification was associated with age-related findings rather than disease status. We found a significant association between age and FLAIR ($P = 0.004$) and MP-RAGE ($P = 0.022$; S4b Fig). There was also a trend for the T2W categorization to be correlated with age ($P = 0.20$). In the younger cohort, 25% of Fabry patients and 16.7% of controls were categorized as cases by FLAIR. The remaining images (79.1%, 19 of 24 participants) appeared normal. In contrast, 11 of 16 (68.8%) older participants were categorized as cases that followed a balanced distribution between Fabry ($N = 5$) and healthy controls ($N = 6$). Overall, the results indicated that qualitative imaging changes in Fabry resembled incidental age-related findings present in healthy controls.

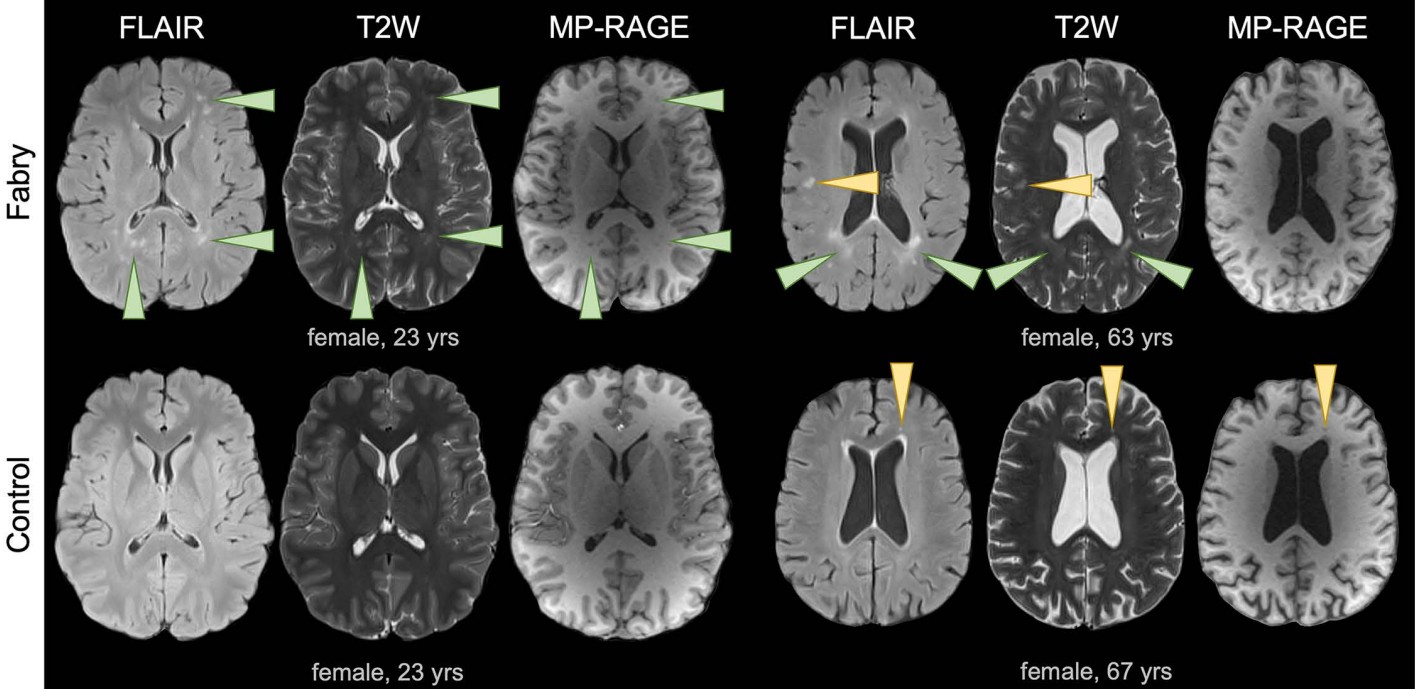

**Fig 2. Large white matter lesions were present in a subset of Fabry participants.** FLAIR, T2W, and MP-RAGE axial images from younger (left) and older (right) Fabry participants are shown to demonstrate the appearance of an increased WML burden in Fabry (top row). The corresponding age/sex-matched controls are also shown (bottom row). Larger WMLs and a confluence of WMLs were evident in Fabry (green arrowheads). Focal lesions (yellow arrowheads) were also present in Fabry but these similarly occurred in controls. Note in both of the older participants (right), enlarged ventricles were present.

During the blinded image review, WMLs were detected with an identical prevalence (60%) in both Fabry and controls (Fisher's test, $P > 0.99$). A strong trend for an increase in the Fazekas scores in Fabry compared to controls was observed ($1.05 \pm 1.05$ vs. $0.5 \pm 0.7$, respectively, $P = 0.059$; Fig 4a). We also found a significant association between age and Fazekas score in Fabry indicating an age-related progression towards greater WML size and lesion confluency which was absent in controls (Fig 4b). In accord, the Fabry regression line slope was significantly different ($F = 5.74$; $P = 0.0014$) compared to the control regression line slope supporting the conjecture that Fabry-related WML progression exceeded normal aging.

**White matter fiber degeneration in Fabry exceeds normal age-related losses**

Brain volumes parallel and superior to the orthogonal plane connecting the genu and splenium of the corpus callosum were extracted using FLAIR and MP-RAGE images (S5 Fig). Voxel counts between Fabry and controls were similar (S6 Fig). Inverse associations between age and brain volume were present and the difference in the slopes of the regression lines between Fabry and controls was not statistically significant (Fig 5a, S7 Fig). The inverse associations between age and brain volume are consistent with normal brain aging [43] and support the robustness of the isolated regions. The volume of the corpus callosum body (CCB) isolated from DTI was inversely correlated with age for Fabry participants but not controls (Fig 5b). To account for age-related changes, CCB volume was normalized (nCCB) using the voxel count from the corresponding FLAIR images. After normalization, the association between nCCB and age persisted in Fabry (Fig 5c). There was a significant difference between the slopes of the regression lines ($F = 7.23$; $P = 0.011$), suggesting an ongoing process in Fabry beyond normal aging. In accord, nCCB volume was similar between groups in the younger cohort, but

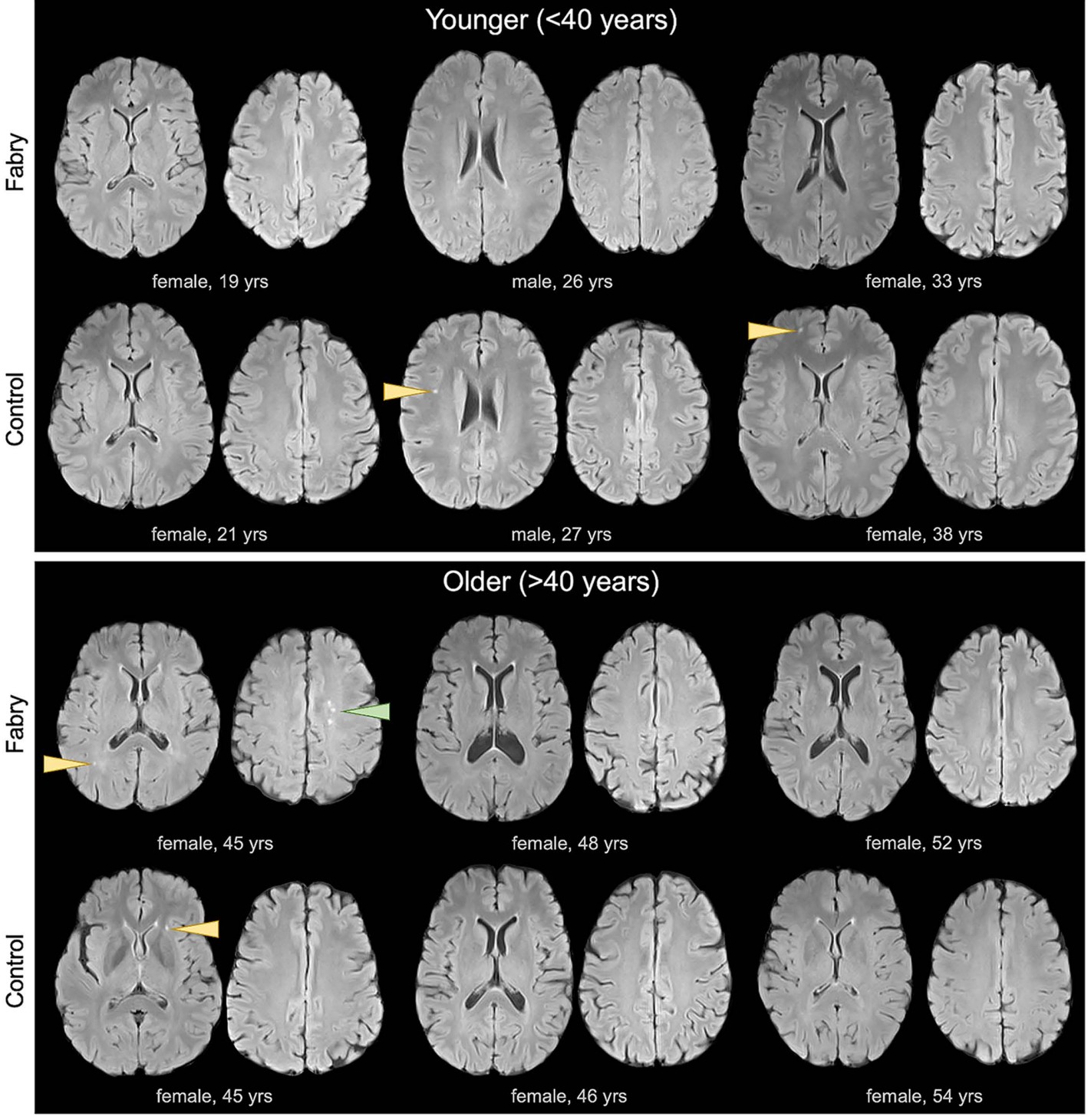

**Fig 3. Punctate white matter lesions are common in Fabry and controls.** FLAIR axial images from Fabry and age/sex-matched controls are shown for three younger (top panel) and three older (bottom panel) participants. Overall, images were largely normal in both cohorts. Punctate WMLs in both Fabry and controls are identified (yellow arrowheads). In one Fabry patient, three punctate WMLs were present in close proximity (bottom panel, green arrowhead).

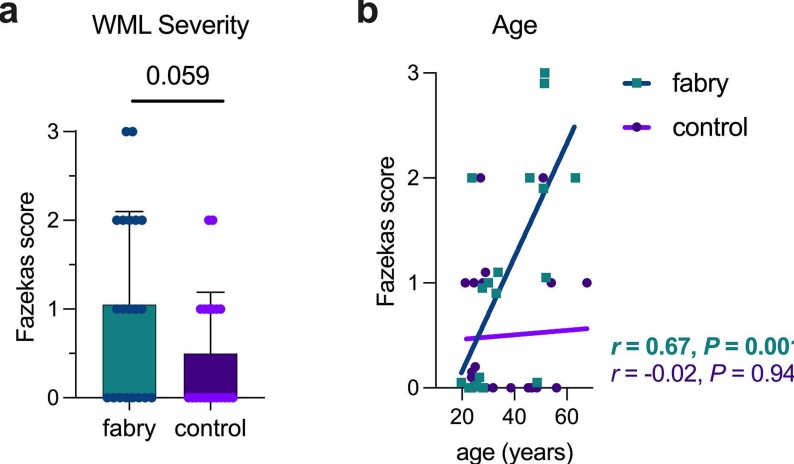

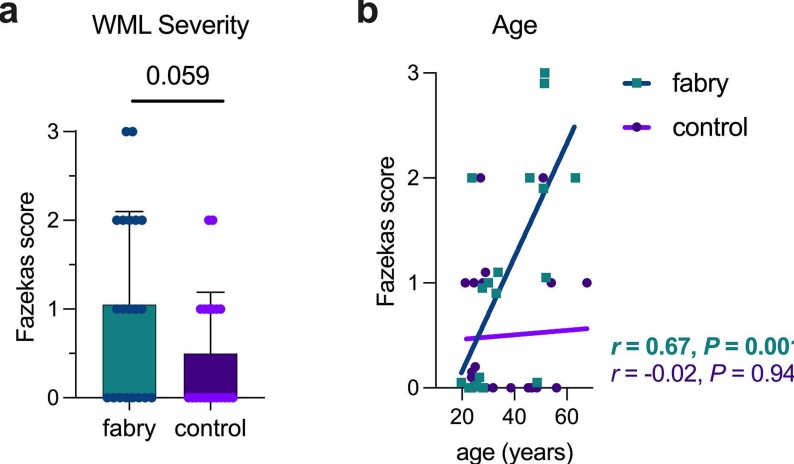

**Fig 4. White matter lesion (WML) severity and age are associated in Fabry.** A trend was present in Fabry for an increased Fazekas score compared to controls (a). In Fabry, a strong correlation between age and Fazekas score was present (b). Note the absence of an association between age and Fazekas score in controls. In (b), data points were nudged from integer values of the Fazekas score to show all data points. Regression lines and correlations were determined from the true data values.

significantly lower for Fabry in the older cohort (Fig 5d). There was not a significant difference in nCCB volume in participants with a WML compared to those without (Fig 5e). A moderate association between nCCB volume and Fazekas score was present in Fabry, but absent in controls (Fig 5f). Thus, the underlying pathophysiology in Fabry affecting brain tissues exhibited a global effect causing WML progression and nCCB volume reduction concomitantly which was in contrast to the unrelated progression of WMLs and nCCB volume in normal aging.

**Microstructural damage to white matter is present in Fabry at a young age**

To evaluate for age-related signal changes, we compared the histograms of normalized signal intensity from controls using the first ($N = 5$; age: 23.7 ± 1.5 yrs) and fourth ($N = 5$; age: 55.5 ± 7.4 yrs) age-based quartiles. A redistribution of normalized signal intensity towards lower values in the older controls was present on FLAIR (Fig 6a), while a shift towards an increase in signal was present on MP-RAGE (S8a Fig). In combination, these differences are consistent with the increased loss of gray matter compared to white matter in normal aging [44]. Subsequently, we restricted histogram comparisons to the younger cohort (<40 yrs) to improve the isolation of Fabry-related changes. An increase in normalized FLAIR intensity was present in Fabry (Fig 6b), which was opposite to that seen with aging. Differences observed between cohorts on MP-RAGE were minimal (S8b Fig). The combined findings from FLAIR and MP-RAGE suggest a mild, diffuse, pathologic process in white matter in the younger Fabry patient cohort.

   Bound-pool fraction imaging (*f*-maps) is a quantitative technique that measures the fraction of hydrogen molecules bound to macromolecules in each voxel. The comparisons of volumes between cohorts (S9a Fig) and age (S9b Fig) were similar to FLAIR. The bimodal distribution of the bound-pool fraction present in younger healthy controls (first quartile) was consistent with previous reports of distinct gray and white matter distributions (Fig 6c) [37,45]. In the older cohort (fourth quartile), gray matter values tended to decrease while white matter values increased, resulting in a more singular central peak consistent with the age-related loss of gray matter (Fig 6c). Similar to FLAIR and MP-RAGE, we restricted the comparison between Fabry and controls to the younger cohort to minimize the effects of age. This was especially important in Fabry patients as the bound-pool fraction is non-specific, and the potential accumulation of the macromolecule lyso-Gb$_3$ in brain tissues with age has an unclear effect on measurements. We observed that the proportion of voxels representing

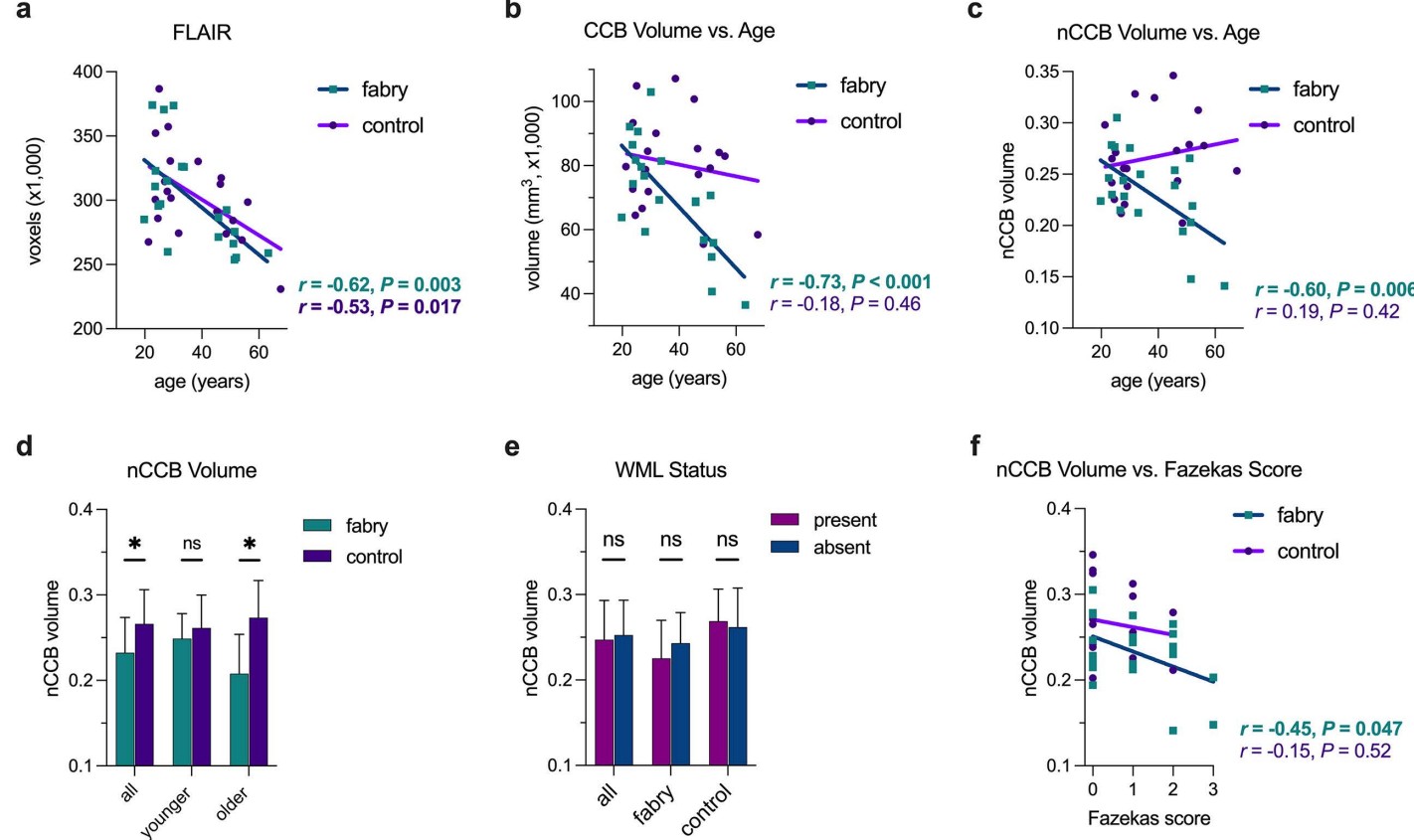

**Fig 5. Fibers associated with the body of the corpus callosum atrophied with age in Fabry.** Associations between age and brain volume (a) and corpus callosum body (CCB) volume (b) are shown for Fabry and controls. In (c), an association between age and CCB persisted for Fabry after normalization (nCCB) for age using the FLAIR brain volumes. In (d), nCCB volume is compared between Fabry and controls. nCCB volume was similar regardless of white matter lesion (WML) status (present vs. absent; e). In (f), associations between nCCB and Fazekas score are shown. Note the wide range of nCCB volumes for both Fabry and controls at a Fazekas score of zero. ns = not significant, *P < 0.05, **P < 0.01, ***P < 0.001.

gray and white matter in the younger Fabry cohort increased and decreased, respectively, indicating reduced myelin density, less overall white matter, or both (Fig 6d).

In the extracted fibers from the corpus callosum body, we found FA (Fig 6e) and MD (Fig 6f) to be associated with age in Fabry and to a lesser extent in controls. There was a strong trend and a significant difference, respectively, between the slopes of the regression lines for FA ($F = 3.62$; $P = 0.065$) and MD ($F = 4.82$; $P = 0.035$) indicating greater age-related alterations to white matter in Fabry. In accord, FA was less (Fig 6g) and MD was greater (Fig 6h) in the Fabry cohort. There was evidence for corresponding differences in the younger Fabry cohort which became statistically significant in the older cohort. As expected, differences and associations in FA and MD were observed based on WML status and Fazekas score (S10 Fig). Our combined observations from FLAIR, f-maps, and DTI provided compelling evidence for the occurrence of microstructural damage to white matter in Fabry at a young age that caused greater damage to white matter over time compared to normal aging alone.

## Associations between imaging metrics and neurocognitive parameters

In both Fabry and controls, verbal IQ was similar regardless of WML status (S11a Fig) and was not associated with Fazekas score (S11b Fig). In Fabry, the absence of a correlation was likely driven, at least in part, by the wide range of verbal

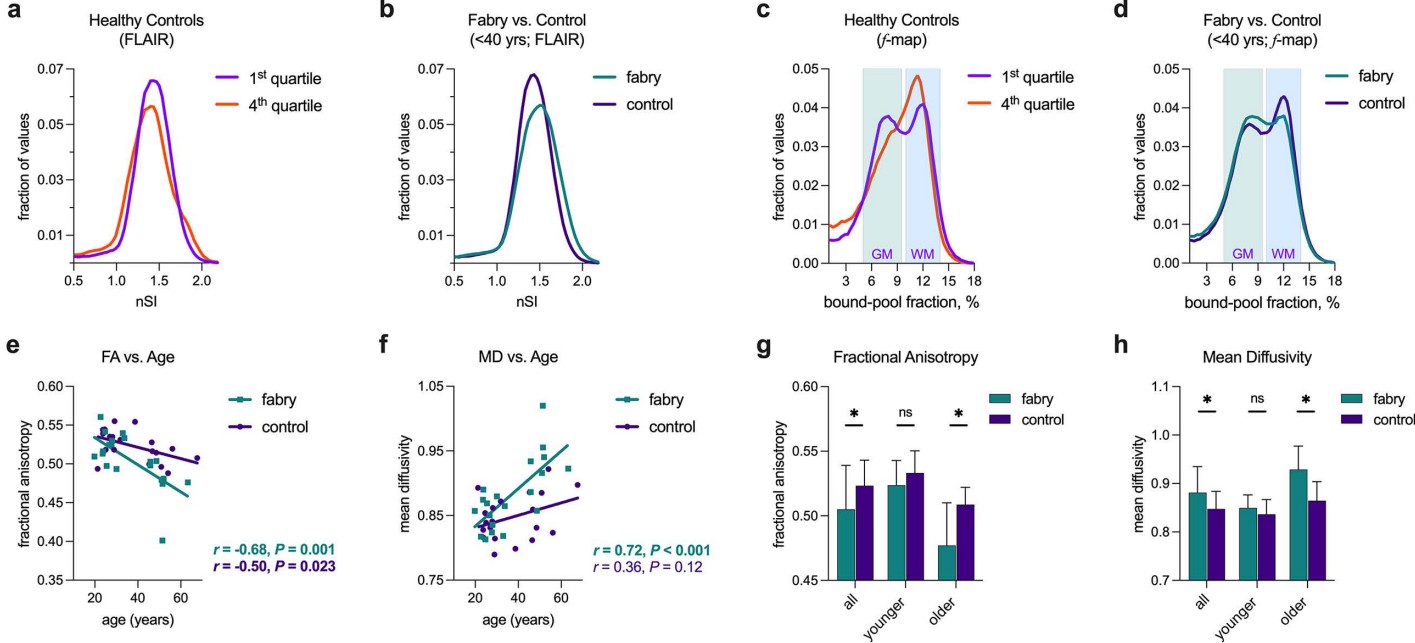

**Fig 6. Early and progressive microstructural damage is present in Fabry disease.** In (a), a comparison of FLAIR histograms using the normalized signal intensity (nSI) from the age-based first and fourth quartiles of healthy controls. The histograms for the FLAIR nSI from the younger adult Fabry and control cohorts are shown in (b). Similar comparisons using histograms from bound-pool fraction maps (*f*-maps) are shown in (c) and (d), respectively. Distribution of values consistent with gray matter (GM) and white matter (WM) are shown. Associations between age and fractional anisotropy (FA; e) and mean diffusivity (MD, f) demonstrate a decrease and increase, respectively in the Fabry cohort with similar trends in controls. Comparisons of fractional anisotropy (g) and mean diffusivity (h) between cohorts shows differences increased in the older cohort. ns = not significant, *$P$ < 0.05, **$P$ < 0.01, ***$P$ < 0.001.

IQ scores in patients with a Fazekas score of zero. Although verbal IQ was associated with the volume of the corpus callosum body (S11c Fig), the significance resolved once the volume data was normalized to age (S11d Fig). Neither FA (S11e Fig) nor MD (S11f Fig) were related to verbal IQ.

Analysis of relationships between presence/absence of error on the Trail Making Test (part A; i.e., executive function) and imaging data were limited to the Fabry cohort because errors were too few in the controls. An association between error on the Trail Making Test (part A) and the presence/absence of a WML (Fisher's test, $P$ = 0.65) was not observed. The Fazekas score, normalized volume of the corpus callosum body, FA, and MD were all similar between Fabry participants with and without an error (S12 Fig). Overall, we did not find a strong association between an imaging metric and neurocognitive findings, particularly after adjusting for age.

## Discussion and conclusions

Our findings support the presence of microstructural damage to white matter in young adults with Fabry. While WMLs were common in both Fabry and controls, the underlying microstructural damage in Fabry may support lesion progression as WML severity in Fabry was age-related, a finding absent in controls. In addition, the underlying pathophysiology affecting Fabry in brain tissues may also lead to the insidious loss of connectivity as reductions in the volume of fibers associated with the corpus callosum were strongly associated with age, a phenomenon that was also absent in controls. Thus, the Fabry-driven accumulation of lyso-Gb$_3$ in brain tissues may lead to the early onset and ongoing microstructural damage of global white matter causing a progressive loss of fiber connections leading to altered cognition that exceeds

the effects of normal aging. Patients with Fabry may benefit from serial neurocognitive studies to identify the onset and evolution of deficits to provide supportive therapies to optimize patient care.

Although the etiology of impaired cognition in Fabry remains unclear, we found evidence that ongoing damage to white matter exceeds normal age-related changes, likely reducing connectivity as evidenced by the observed atrophy of the corpus callosum. An abnormal corpus callosum has been previously implicated as a source of reduced cognition. A comparison of 26 individuals with normal intelligence and complete or partial agenesis of the corpus callosum compared to 24 matched controls found the former group exhibited intact attention and effectively retained and retrieved previously learned information, but learned less during verbal encoding [46]. While these findings identify the role of the corpus callosum in the facilitation of interhemispheric integration, the study also highlights the relatively subtle effects due to brain plasticity that may occur with alterations to the corpus callosum even in the context of complete agenesis [46,47]. More relevant to the findings described herein, are previous reports of acquired alterations to the corpus callosum associated with cognitive decline. A study of lead exposure in children found smaller volumes of the corpus callosum were linked to poorer performance on cognitive tests measuring language and processing speed [48]. In 22 adults with multiple sclerosis followed serially with MRI for nine years, callosal atrophy was related to cognitive speed [49]. Similarly, in 21 patients clinically diagnosed with Alzheimer's disease, the atrophy rate of the corpus callosum was correlated with the progression of dementia severity [50]. Notably, single time point measurements of the corpus callosum have emerged as indicators of cognitive decline in multiple sclerosis [51], Alzheimer's disease [52], and Parkinson's disease [53]. Our findings deviate from these prior studies as we did not find alterations to the corpus callosum body alone governed the decline in verbal acuity and executive function in Fabry. The absence of an association between the corpus callosum and cognitive decline may be due to the relatively early-onset and insidious effect of Fabry on white matter compared to other acute conditions or neurodegenerative conditions that tend to occur later in life. Similarly, we also did not find the presence of WMLs and WML severity to be associated with cognitive decline in contrast to prior reports of late-onset neurodegenerative conditions such as Alzheimer's [54] and Parkinson's [55] disease. Notably, more recent long-term studies of dementia have found that generalized white matter atrophy was an important risk factor for cognitive impairment [56]. Thus, our data indicate that Fabry-related alterations may be global and the complex and inefficient connectivity required to perform verbal-based tasks and executive functions may be most susceptible to the generalized degradation of connections [57]. Future studies may benefit from serial measurements of the corpus callosum in Fabry to better define and discern the effects of atrophy that may vary between individuals.

The leveraging of multiple imaging markers in combination with a comparison to the effects of normal brain aging enabled our detection of microstructural damage to white matter in the younger Fabry cohort. Specifically, we found (1) increased normalized signal intensity on FLAIR, (2) an absence of differences on MP-RAGE, (3) reduced representation of high-density white matter on bound-pool fraction imaging, and (4) reduced FA and increased MD on DTI. Because of age-related effects, we were unable to study the normalized signal intensity of FLAIR, MP-RAGE, and bound-pool fraction imaging in the older cohort. We note that using bound-pool fraction imaging in the older cohort may be particularly problematic in Fabry due to the accumulation of lyso-Gb$_3$, a macromolecule whose rich hydrogen content may mitigate the loss of myelin-associated macromolecules, confounding data interpretation. Diffusion-based metrics, however, enabled the determination that the trends in FA and MD detected in the younger cohort amplified with age and exceeded normal brain aging, supporting ongoing damage to white matter, which is consistent with observations by others that alterations to diffusion-based parameters preceded the development of WMLs [22]. The source of white matter damage in Fabry is unclear. Although autopsy studies have uniformly identified an abnormal accumulation of glycolipids in brain tissues, the affected cells and location of the accumulation vary widely. For example, Tabira et al. described glycolipid deposition in neurons and glial cells, while Lou and Reske-Nielsen reported that abnormal glycolipid deposits were absent in both of these cell types [58,59]. Instead, they found pathological changes were dominant in all cerebral vessels (i.e., arteries to capillaries and veins), which included deposition of glycolipids in smooth muscle cells and endothelial cells [59]. The most

recent autopsy study from Okeda et al. in 2008 found a sparse presence of neuronal ballooning, more prominent astro-cytic swelling, accumulation of glycolipid in medial smooth muscle cells limited to the subarachnoid arteries, and absence of glycolipid deposits in both endothelial cells and intraparenchymal cortical and medullary arteries of the cerebrum [60]. Thus, the heterogeneous sites of lyso-Gb$_3$ accumulation in the cerebrum may be patient-specific, which makes establishing a common mechanism that governs microstructural damage challenging. While possibilities have been proposed [60], the identification of a mechanism for microstructural damage to white matter and other neurologic sequelae remains elusive. Regardless, the underlying etiology appears to have an insidious effect on both white matter and cognition that may not manifest clinically for decades.

Our findings occurred in a majority female Fabry cohort. Although Fabry is an X-linked disease, the recognition that heterozygous females similarly experience the same significant manifestations of the disease as hemizygous males has been increasingly appreciated since the turn of the century [61]. Our study further supports female susceptibility to developing the diverse neurologic sequelae of Fabry. We also note that all Fabry participants were stroke-free and nearly all reported being on ERT. Although we did not collect data specific to the duration, compliance, and biomarkers (e.g., serum globotriaosylsphingosine level) of ERT, our data support the occurrence of cognitive decline in the absence of stroke, and the possibility of neurologic complications regardless of ERT status. Our findings are supported by prior studies that did not find compelling evidence ERT profoundly mitigates the neurologic sequelae of Fabry. Rombach et al. showed ERT did not affect the development of first stroke in 100 Fabry patients [5]. Anderson et al. found the risk of stroke before the age of 40 was low regardless of ERT status and also observed no significant differences in having a TIA/stroke on or off therapy [62]. There has been evidence, however, that duration of ERT may reduce the risk of stroke. For example, in a cohort of 1,044 Fabry patients receiving agalsidase beta for a mean period of 2.8 years started at a median age of 40 years, severe clinical events, including stroke were reduced once treatment duration extended beyond six months [63]. Rombach et al. also described evidence that ERT duration >4.2 years reduced major first complications (i.e., cardiac event, end-stage renal disease, stroke, and death). In contrast, Sirrs et al. studied 86 patients previously on ERT for a mean of ~34 months and 92 patients naive to ERT randomized to either agalsidase alfa or agalsidase beta and found no difference in acute neurological event rates between the two cohorts with a median follow-up of 64 months and 59 months, respectively, suggesting that intervention earlier in the course of Fabry with ERT did not reduce stroke risk [64]. Beyond stroke, the effects of ERT also do not seem to impact WMLs. In the study by Rombach et al., 48% of males and 28% of females developed new WMLs while on ERT for a median duration of 3.1 years and 4.0 years, respectively [5]. Fellgiebel et al. studied 41 patients with Fabry (25 on ERT; 16 on placebo) with an average follow-up of 27 months and found WML burden increased, but the proportion of patients with a stable or decreased WML load was significantly higher in the ERT group (44%, 8 of 18) compared to placebo (31%, 4 of 13) [17]. In a meta-analysis of 199 Fabry patients, Korver et al. reported WMLs progressed in 24.6% of patients regardless of ERT status during a mean follow-up of 38.1 months [9]. In a review of 863 brain MRIs from 149 patients, lesion progression was independent of ERT status [65]. The nominal effect of current therapies on neurologic manifestations of Fabry is likely attributable to the inability of ERT to penetrate the blood-brain barrier. Thus, newer therapies capable of crossing the blood-brain barrier or reducing lyso-Gb$_3$ in the brain may have stronger efficacy, and future neurocognitive studies of patients on short- and long-term therapy are warranted [66,67].

We found verbal IQ and executive function to be the most adversely affected cognitive parameters in our Fabry cohort. A variety of neurocognitive deficits identified in Fabry, including verbal acumen and executive function, have been extensively reviewed in prior works [11,15,19,68]. The differences in specific deficits between studies are most likely attributable to relatively small sample sizes in combination with the nonuniform testing paradigms between investigations. Nevertheless, there is a collective abundance of evidence that neurocognitive deficits manifest in Fabry. Our data indicates that altered cognition develops over time and rate of decline exceeds normal age-related changes. Although we found imaging metrics that similarly evolved with age (e.g., volume of the corpus callosum, mean diffusivity, etc.), a correlation between an imaging metric and altered cognition was absent after accounting for age and

an underlying etiology for the decline was not established. Because the neurological effects of Fabry develop over decades, neural plasticity may account for the absence of a specific imaging surrogate and the relatively subtle deficits in Fabry compared to more acute processes such as multiple sclerosis which similarly develops WMLs [69]. As such, serial MRIs to monitor the neurological sequelae of Fabry may not be clinically impactful in the absence of concerns for stroke/TIA. Current guidelines recommend a baseline brain MRI in male patients over 21 years and female patients over 30 years with a monitoring schedule of at least once every three years [70,71]. A baseline MRI for future comparisons to new onset TIA/stroke-like symptoms seems clinically warranted. As an alternative to serial MRIs in the absence of TIA/stroke, however, we propose using serial neurocognitive studies to monitor patients for cognitive decline since an imaging parameter alone is unlikely to identify a patient-specific deficit. Future investigations that include serial neurocognitive studies to identify the rate and effects of therapies on decline may prove impactful in establishing and monitoring treatment strategies for Fabry disease.

We note several limitations to our study. First, the small sample size may be impacted by selection bias. Not only by the recruitment of healthy controls from a university campus, as identified earlier, but the reliance on Fabry participants to volunteer may have also affected RAND-36 parameters, such as physical functioning, role limitations, pain, etc. Second, we limited our study to a single, large supratentorial white matter fiber tract because of the small sample size to mitigate Type I error. Thus, we did not ascertain if other fiber tracts may be more or less susceptible to the effects of Fabry disease. Future studies using a larger sample size in combination with a higher spatial resolution may enable the analysis of the full complement of fiber tracts to uncover distinct pathways affected by Fabry disease, such as that recently shown in Alzheimer's disease [72]. Finally, our semi-quantitative and quantitative imaging approach detected differences between cohorts, but the variations we observed were subtle and likely occurred over years to decades. As such, the study did not yield a strong candidate for an imaging biomarker that may be sufficiently dynamic for monitoring during clinical trials seeking to study the effects of current or new Fabry therapies on the brain. Nevertheless, the integration of quantitative and semi-quantitative imaging techniques into natural history studies may prove valuable in better understanding the long-term neurologic sequelae of Fabry disease.

## Supporting information

**S1 Fig. Demographics and clinical characteristics of the case-control cohorts.** Age (a), sex (b), and ethnicity (c) are compared between participants with Fabry disease and controls. In (d), clinical characteristics are compared. htn = hypertension, chol = hypercholesterolemia, ns = not significant.
(PDF)

**S2 Fig. Association between age and verbal intelligence quotient (IQ).** In (a), a scatter plot of verbal IQ vs. age is shown along with regression lines. Although verbal IQ tended to decrease with age in both cohorts, the association was not significant (Pearson's *r*). For Fabry (b) and controls (c), a range of verbal IQ values were present regardless of age and sex. Bars represent median values.
(PDF)

**S3 Fig. Years of education in Fabry participants with an error during the Trail Making Test (Part A).** Years of education were similar between Fabry patients with and without an error during the Trail Making Test (Part A). ns = not significant.
(PDF)

**S4 Fig. Contingency tables for case/control categorization by qualitative image interpretation.** In (a), tables are shown for the blinded classification of case/control status by the study's neuroradiologist using FLAIR, T2W, and MP-RAGE images. In (b), the same classification by the study's neuroradiologist is used but the truth set was regrouped

based on age (younger, <40 yrs vs. older >40 yrs) rather than case/control status. In (a) and (b), *P*-values obtained using Fisher's exact test are shown under each table.
(PDF)

**S5 Fig. Brain regions selected for histogram analysis.** Representative reconstructed sagittal images from FLAIR (left) and MP-RAGE (right) images are shown off-center from the corpus callosum to provide a representation of brain tissues sampled. The dashed gray line outlines the field-of-view for each image. The shaded region superior to the solid gray line (left) and between the two solid gray lines (right) were isolated for histogram and volume analysis. Images are shown before the brain tissue mask was applied to show anatomic detail for reference.
(PDF)

**S6 Fig. Comparison of structural MRI brain volumes. ns = not significant.**
(PDF)

**S7 Fig. Association between brain volume and age using MP-RAGE.** Although Pearson's correlation (*r*) was significant in the Fabry cohort, the controls followed a similar trend, and the difference between the slopes of the regression lines was not significant.
(PDF)

**S8 Fig. Comparison of histograms using MP-RAGE.** In (a), a comparison of MP-RAGE histograms using the normalized signal intensity (nSI) from the age-based first and fourth quartiles of healthy controls. The histograms for the MP-RAGE nSI from the younger adult (<40 yrs) Fabry and control cohorts are shown in (b).
(PDF)

**S9 Fig. Bound-pool fraction volumes and association with age.** In (a), the comparison of volumes between Fabry and controls is shown. In (b), the association between brain volumes and age is shown. Although Pearson's correlation (*r*) was significant in controls, the Fabry cohort followed a similar trend. ns = not significant.
(PDF)

**S10 Fig. White matter lesions (WMLs) and effects on diffusion-based parameters.** Fractional anisotropy (a) and mean diffusivity (b) are compared based on presence/absence of WMLs. Associations between Fazekas score and fractional anisotropy (FA; c) and mean diffusivity (MD, d) show significant correlations in Fabry and strong trends in controls.
(PDF)

**S11 Fig. Comparisons between verbal IQ and imaging metrics.** In (a), verbal IQ is compared between the presence/absence of a white matter lesion (WML). The association between verbal IQ and Fazekas score is shown in (b). Although a significant association (Pearson's *r*) between verbal IQ and corpus callosum body (CCB) volume was present in Fabry (c), the significance resolved once the volume was normalized (nCCB) for age (d). Significant associations were not observed between age and fractional anisotropy (e) and mean diffusivity (f).
(PDF)

**S12 Fig. Comparisons between imaging metrics and executive function in the Fabry cohort.** Fazekas score (**a**), normalized corpus callosum body (nCCB) volume (**b**), fractional anisotropy (**c**), and mean diffusivity (**d**) are compared in the Fabry cohort based on the presence/absence of error during the Trail Making Test (Part A). ns = not significant.
(PDF)

**S1 File. Supplemental Data File.**
(XLSX)

## Author contributions

**Conceptualization:** Brandon A. Zielinski, Hunter R. Underhill.

**Data curation:** Jacob W. Johnson, Brandon A. Zielinski, Hunter R. Underhill.

**Formal analysis:** Jacob W. Johnson, Brandon A. Zielinski, Hunter R. Underhill.

**Funding acquisition:** Brandon A. Zielinski, Hunter R. Underhill.

**Investigation:** Jacob W. Johnson, Hediyeh Baradaran, Jubel Morgan, Henrik Odèen, Emma Friel, Carrie Bailey, Brandon A. Zielinski, Hunter R. Underhill.

**Methodology:** Jacob W. Johnson, Hediyeh Baradaran, Jubel Morgan, Henrik Odèen, Emma Friel, Brandon A. Zielinski, Hunter R. Underhill.

**Project administration:** Carrie Bailey, Brandon A. Zielinski, Hunter R. Underhill.

**Resources:** Hediyeh Baradaran, Jubel Morgan, Carrie Bailey, Brandon A. Zielinski, Hunter R. Underhill.

**Software:** Jacob W. Johnson, Henrik Odèen, Brandon A. Zielinski, Hunter R. Underhill.

**Supervision:** Brandon A. Zielinski, Hunter R. Underhill.

**Validation:** Hunter R. Underhill.

**Visualization:** Hunter R. Underhill.

**Writing – original draft:** Hediyeh Baradaran, Hunter R. Underhill.

**Writing – review & editing:** Jacob W. Johnson, Hediyeh Baradaran, Jubel Morgan, Henrik Odèen, Emma Friel, Carrie Bailey, Brandon A. Zielinski, Hunter R. Underhill.

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
