## [Decision Letter · Decision Letter 0]

18 Jun 2025

Dear Dr. Underhill,

Thank you for submitting your manuscript to PLOS ONE. After careful consideration, we feel that it has merit but does not fully meet PLOS ONE’s publication criteria as it currently stands. Therefore, we invite you to submit a revised version of the manuscript that addresses the points raised during the review process.

We look forward to receiving your revised manuscript.

Kind regards,

Maria de Fátima Macedo, Ph.D.

Academic Editor

PLOS ONE

2. In the online submission form, you indicated that DICOM files are available from the corresponding author upon reasonable request and following approval from the study group.

3. Please expand the acronym “NIH” (as indicated in your financial disclosure) so that it states the name of your funders in full.

Genzyme, a Sanofi corporation supported this investigator sponsored study (GZ-2015-11321). The MRI resources used were partially funded by an NIH Shared Instrumentation Grant (S10OD018482).

Reviewers' comments:

Reviewer's Responses to Questions

**Comments to the Author**

1. Is the manuscript technically sound, and do the data support the conclusions?

Reviewer #1: Yes

Reviewer #2: Yes

2. Has the statistical analysis been performed appropriately and rigorously?

Reviewer #1: Yes

Reviewer #2: Yes

3. Have the authors made all data underlying the findings in their manuscript fully available?

Reviewer #1: Yes

Reviewer #2: No

4. Is the manuscript presented in an intelligible fashion and written in standard English?

Reviewer #1: Yes

Reviewer #2: Yes

Reviewer #1: This study investigates cognitive decline and white matter damage in Fabry disease using advanced neuroimaging. Despite similar qualitative MRI findings to healthy aging, quantitative metrics (e.g., DTI, bound-pool imaging) revealed early microstructural damage in Fabry patients. DTI showed decreased fractional anisotropy (FA) and increased mean diffusivity (MD), particularly in the corpus callosum. Cognitive assessments indicated reduced verbal IQ and executive function in Fabry patients, worsening with age. The study suggests microstructural white matter degeneration precedes visible lesions and contributes to cognitive decline.

Interpretation of DTI Metrics: The study highlights FA and MD changes in the corpus callosum as evidence of early microstructural damage in Fabry disease. However, could the authors clarify whether these findings are regionally specific or represent a diffuse pattern across other white matter tracts as well?

Bound-Pool Fraction Imaging and Lyso-Gb3: Given that bound-pool fraction imaging may be influenced by macromolecule accumulation such as lyso-Gb3, how do the authors ensure that the observed signal alterations are due to myelin changes rather than Fabry-specific glycolipid deposition?

Clinical Translation and Longitudinal Value: While serial neurocognitive testing is proposed, do the authors believe any of the imaging markers (e.g., nCCB volume, FA, MD) could serve as reliable surrogate endpoints in future clinical trials assessing CNS-targeted therapies for Fabry disease?

Recommended Citation: Several previous studies have investigated the longitudinal impact of white matter atrophy on cognitive function, as well as the influence of more specific hippocampal tracts on cognition. Please consider incorporating these findings into the Discussion section, with reference to the examples below:

10.1001/jamanetworkopen.2024.41505

10.1002/alz.70142

Reviewer #2: 1. "Neurocognitive assessments identified trends for lower verbal intelligence quotient and executive function in the younger Fabry participants, which became statistically significant in the older Fabry cohort." Please replace “older Fabry cohort” by “older Fabry patients”, as this is a case-control study.

2. "Our data indicate that the early onset of microstructural damage in Fabry drives the insidious degeneration of white matter, leading to reduced cognition." replace “indicate” by “suggest” which more coherent with a case-control study showing associations

3. In the methods: Include a description of the conditions under which cognitive evaluations were conducted, including scheduling, duration, and whether assessments were performed in person or remotely.

4. Line 296: "A board-certified neuroradiologist blinded..." This description of the evaluation process should be moved to the Methods section.

5. Line 348: "In combination, these differences are consistent with the increased loss of gray matter compared to white matter in normal aging [44] and demonstrate the usefulness of histogram analysis to detect relatively subtle perturbations between cohorts." This interpretation should be moved to the discussion section, not presented within the results.

6. Although the etiology of reduced cognition in Fabry remains unclear". Please replace reduced cognition with impaired cognition.

7. The Discussion should include a section on study limitations, such as sample size and potential selection bias.

8. This study was partially sponsored by Genzyme – a description of the industry role (or its absence) should be cleared

**Do you want your identity to be public for this peer review?** For information about this choice, including consent withdrawal, please see our Privacy Policy

Reviewer #1: **Yes: ** Yuto Uchida

Reviewer #2: No

---

## [Author Response · Author response to Decision Letter 1]

19 Sep 2025

We thank the reviewers and PLOS ONE for your time and thoughtful comments. We have revised the manuscript accordingly and believe the revisions strengthen our findings. The following is a point-by-point response (regular font) to each comment/criticism (blue italic font).

The author page and manuscript have been revised to meet PLOS ONE’s style requirements.

2. In the online submission form, you indicated that DICOM files are available from the corresponding author upon reasonable request and following approval from the study group.

This policy applies to all data except where public deposition would breach compliance with the protocol approved by your research ethics board. If your data cannot be made publicly available for ethical or legal reasons (e.g., public availability would compromise patient privacy), please explain your reasons on resubmission, and your exemption request will be escalated for approval.

In the revised manuscript, we have added a data file used to generate the figures/tables in the manuscript (S1 File). All magnetic resonance images used in the study have been uploaded to the Dryad Data Repository and will be made available after acceptance of the manuscript for publication. The following link from Dryad can be used to confirm the data upload/availability:

http://datadryad.org/share/LINK_NOT_FOR_PUBLICATION/CVHUdLlDW9-ritEdqL1Bm9DVxeEmtvsVJSrUZ-nTcw4

For publication, the following Dryad link identifies the dataset: https://doi.org/10.5061/dryad.gb5mkkx3g

We changed our data availability statement to the following:

“All data not contained in the paper necessary to replicate plots and figures are available in the S1 file. All magnetic resonance images (DICOM, RAW) used in the study are available from the Dryad Data Repository (https://doi.org/10.5061/dryad.gb5mkkx3g).”

3. Please expand the acronym “NIH” (as indicated in your financial disclosure) so that it states the name of your funders in full. This information should be included in your cover letter; we will change the online submission form on your behalf.

We have expanded the acronym “NIH” to “National Institutes of Health” in our cover letter as shown in our response to #4.

Genzyme, a Sanofi corporation supported this investigator sponsored study (GZ-2015-11321). The MRI resources used were partially funded by an NIH Shared Instrumentation Grant (S10OD018482).

We have included the following statement in our revised cover letter:

“Genzyme, a Sanofi corporation, supported this investigator sponsored study (GZ-2015-11321). The magnetic resonance imaging resources used were partially funded by a National Institutes of Health Shared Instrumentation Grant (S10OD018482). The funders had no role in study design, data collection and analysis, decision to publish, or preparation of the manuscript.”

Captions for the Supporting Information files have been placed at the end of the revised manuscript, and any in-text citations have been matched accordingly.

Reviewers' comments:

Reviewer's Responses to Questions

Comments to the Author

1. Is the manuscript technically sound, and do the data support the conclusions?

Reviewer #1: Yes

Reviewer #2: Yes

2. Has the statistical analysis been performed appropriately and rigorously?

Reviewer #1: Yes

Reviewer #2: Yes

3. Have the authors made all data underlying the findings in their manuscript fully available?

Reviewer #1: Yes

Reviewer #2: No

In the revised manuscript, we have added a data file used to generate the figures/tables in the manuscript (S1 File). All magnetic resonance images used in the study have been uploaded to the Dryad Data Repository and will be made available after acceptance of the manuscript for publication. The following link from Dryad can be used to confirm the data upload/availability:

http://datadryad.org/share/LINK_NOT_FOR_PUBLICATION/CVHUdLlDW9-ritEdqL1Bm9DVxeEmtvsVJSrUZ-nTcw4

For publication, the following Dryad link identifies the dataset: https://doi.org/10.5061/dryad.gb5mkkx3g

In the revised manuscript at Line 233, we changed our data availability statement to the following:

“All data not contained in the paper necessary to replicate plots and figures are available in the S1 file. All magnetic resonance images (DICOM, RAW) used in the study are available from the Dryad Data Repository (https://doi.org/10.5061/dryad.gb5mkkx3g).”

4. Is the manuscript presented in an intelligible fashion and written in standard English?

Reviewer #1: Yes

Reviewer #2: Yes

5. Review Comments to the Author

Reviewer #1: This study investigates cognitive decline and white matter damage in Fabry disease using advanced neuroimaging. Despite similar qualitative MRI findings to healthy aging, quantitative metrics (e.g., DTI, bound-pool imaging) revealed early microstructural damage in Fabry patients. DTI showed decreased fractional anisotropy (FA) and increased mean diffusivity (MD), particularly in the corpus callosum. Cognitive assessments indicated reduced verbal IQ and executive function in Fabry patients, worsening with age. The study suggests microstructural white matter degeneration precedes visible lesions and contributes to cognitive decline.

Interpretation of DTI Metrics: The study highlights FA and MD changes in the corpus callosum as evidence of early microstructural damage in Fabry disease. However, could the authors clarify whether these findings are regionally specific or represent a diffuse pattern across other white matter tracts as well?

Because of the small sample size, we limited our analysis to the largest supratentorial fiber tract and did not investigate other fiber tracts because of statistical concerns (Type I error). In the revised manuscript, a limitations paragraph was added, and the following text was included at Line 575:

“Second, we limited our study to a single, large supratentorial white matter fiber tract because of the small sample size to mitigate Type I error. Thus, we did not ascertain if other fiber tracts may be more or less susceptible to the effects of Fabry disease. Future studies using a larger sample size in combination with a higher spatial resolution may enable the analysis of the full complement of fiber tracts to uncover distinct pathways affected by Fabry disease, such as that recently shown in Alzheimer’s disease [72].”

Bound-Pool Fraction Imaging and Lyso-Gb3: Given that bound-pool fraction imaging may be influenced by macromolecule accumulation such as lyso-Gb3, how do the authors ensure that the observed signal alterations are due to myelin changes rather than Fabry-specific glycolipid deposition?

We agree that the accumulation of Lyso-Gb3 could be a confounder. Our rationale that the observed signal alterations are most likely due to myelin changes is that Lyso-Gb3 is a macromolecule, and an accumulation would increase rather than decrease bound-pool fraction measurements. Studying younger individuals with Fabry disease is compelling because we expect the accumulation of Lyso-Gb3 in endothelial cells to mitigate some of the signal loss caused by a reduction in myelin density.

In the revised manuscript, the text includes the following statement at Line 487:

“Because of age-related effects, we were unable to study the normalized signal intensity of FLAIR, MP-RAGE, and bound-pool fraction imaging in the older cohort. We note that using bound-pool fraction imaging in the older cohort may be particularly problematic in Fabry due to the accumulation of lyso-Gb3, a macromolecule whose rich hydrogen content may mitigate the loss of myelin-associated macromolecules, confounding data interpretation.”

Clinical Translation and Longitudinal Value: While serial neurocognitive testing is proposed, do the authors believe any of the imaging markers (e.g., nCCB volume, FA, MD) could serve as reliable surrogate endpoints in future clinical trials assessing CNS-targeted therapies for Fabry disease?

Although differences were observed between Fabry patients and controls in the younger cohort, the rate of change seems to occur over decades, likely making the imaging markers not suitable for a clinical trial with a narrow study period. In the revised manuscript, we have included a limitations paragraph with the following text at Line 581:

“Finally, our semi-quantitative and quantitative imaging approach detected differences between cohorts, but the variations we observed were subtle and likely occurred over years to decades. As such, the study did not yield a strong candidate for an imaging biomarker that may be sufficiently dynamic for monitoring during clinical trials seeking to study the effects of current or new Fabry therapies on the brain. Nevertheless, the integration of quantitative and semi-quantitative imaging techniques into natural history studies may prove valuable in better understanding the long-term neurologic sequelae of Fabry disease.”

Recommended Citation: Several previous studies have investigated the longitudinal impact of white matter atrophy on cognitive function, as well as the influence of more specific hippocampal tracts on cognition. Please consider incorporating these findings into the Discussion section, with reference to the examples below:

10.1001/jamanetworkopen.2024.41505

10.1002/alz.70142

We appreciate these references and have added them to the revised manuscript. We have added the following text to the Discussion:

Line 476: “Notably, more recent long-term studies of dementia have found that generalized white matter atrophy was an important risk factor for cognitive impairment [56]”

Line 578: “Future studies using a larger sample size in combination with a higher spatial resolution may enable the analysis of the full complement of fiber tracts to uncover distinct pathways affected by Fabry disease, such as that recently shown in Alzheimer’s disease [72]”

Reviewer #2:

1. "Neurocognitive assessments identified trends for lower verbal intelligence quotient and executive function in the younger Fabry participants, which became statistically significant in the older Fabry cohort." Please replace “older Fabry cohort” by “older Fabry patients”, as this is a case-control study.

In Line 40 of the revised manuscript, we have replaced “older Fabry cohort” with “older Fabry patients.”

2. "Our data indicate that the early onset of microstructural damage in Fabry drives the insidious degeneration of white matter, leading to reduced cognition." replace “indicate” by “suggest” which is more coherent with a case-control study showing associations.

In Line 40 of the revised manuscript, we have replaced “indicate” with “suggest.”.

3. In the methods: Include a description of the conditions under which cognitive evaluations were conducted, including scheduling, duration, and whether assessments were performed in person or remotely.

At Line 136 in the revised Methods, we have added/edited the following text:

“The questionnaires and neurocognitive evaluations occurred within two weeks of the participant’s corresponding MRI, were conducted in person by the study nurse with >20 years of experience administering neuropsychological assessments, and the assessment duration was <4 hours.”

4. Line 296: "A board-certified neuroradiologist blinded..." This description of the evaluation process should be moved to the Methods section.

We have removed this description in the Results. At Line 192 in the Methods section, a description of the evaluation process is provided:

“A board-certified neuroradiologist with a certificate of added qualification in neuroradiology with over 10 years of experience performed all image review blinded to age and case-control status. During the initial review, the study neuroradiologist was shown FLAIR images in the axial plane from all participants in random order to categorize case-control status. An identical process was performed for T2W and MP-RAGE images.”

The revised text in the Results at Line 295 now reads:

“Using FLAIR images, the study neuroradiologist was unable to correctly assign case/control status (Fisher’s test, P>0.99; S4a Fig).”

5. Line 348: "In combination, these differences are consistent with the increased loss of gray matter compared to white matter in normal aging [44] and demonstrate the usefulness of histogram analysis to detect relatively subtle perturbations between cohorts." This interpretation should be moved to the discussion section, not presented within the results.

In the revised manuscript, we have parsed this sentence. The first part was left in place in the revised manuscript because the statement supports the transition of studying young participants to mitigate age-related effects. Thus, the revised section at Line 377 now reads:

“In combination, these differences are consistent with the increased loss of gray matter compared to white matter in normal aging [44]. Subsequently, we restricted histogram comparisons to the younger cohort (<40 yrs) to improve the isolation of Fabry-related changes.”

The second part was largely removed from the revised manuscript, although the intent was rephrased and added to the final paragraph of the Discussion at Line 581:

“Finally, our semi-quantitative and quantitative imaging approach detected differences between cohorts, but the var

---

## [Decision Letter · Decision Letter 1]

6 Oct 2025

The insidious degeneration of white matter and cognitive decline in Fabry disease

PONE-D-25-23651R1

Dear Dr. Hunter R Underhill,

We’re pleased to inform you that your manuscript has been judged scientifically suitable for publication and will be formally accepted for publication once it meets all outstanding technical requirements.

Kind regards,

Maria de Fátima Matos Almeida Henriques de Macedo, Ph.D.

Academic Editor

PLOS ONE

Additional Editor Comments (optional):

Reviewers' comments:

Reviewer's Responses to Questions

**Comments to the Author**

Reviewer #1: All comments have been addressed

Reviewer #2: All comments have been addressed

2. Is the manuscript technically sound, and do the data support the conclusions?

Reviewer #1: Yes

Reviewer #2: Yes

3. Has the statistical analysis been performed appropriately and rigorously?

Reviewer #1: Yes

Reviewer #2: Yes

4. Have the authors made all data underlying the findings in their manuscript fully available?

Reviewer #1: Yes

Reviewer #2: Yes

5. Is the manuscript presented in an intelligible fashion and written in standard English?

Reviewer #1: Yes

Reviewer #2: Yes

Reviewer #1: The authors have properly addressed all of my previous concerns. The revised manuscript is clearly improved, and the additional explanations and analyses strengthen the conclusions. I have no further major comments, and I believe the manuscript is now suitable for publication.

Reviewer #2: The authors have responded to all my comments. I do not have any further points. The manuscript is now suitable for publication.

**Do you want your identity to be public for this peer review?** For information about this choice, including consent withdrawal, please see our Privacy Policy

Reviewer #1: **Yes: ** Yuto Uchida

Reviewer #2: No

---

## [Editor Report · Acceptance letter]

PONE-D-25-23651R1

PLOS ONE

Dear Dr. Underhill,

I'm pleased to inform you that your manuscript has been deemed suitable for publication in PLOS ONE. Congratulations! Your manuscript is now being handed over to our production team.

Kind regards,

on behalf of

Prof. Maria de Fátima Matos Almeida Henriques de Macedo

Academic Editor

PLOS ONE